# Association between Locomotive Syndrome and Physical Activity in Long-Term Inpatients of Psychiatric Care Wards in Japan: A Preliminary Study

**DOI:** 10.3390/healthcare10091741

**Published:** 2022-09-11

**Authors:** Yusuke Ishibashi, Muneyoshi Nishida, Motoharu Hirai, Sae Uezono, Sosuke Kitakaze, Munetsugu Kota, Yukihide Nishimura, Fumihiro Tajima, Hideki Arakawa

**Affiliations:** 1Akitsu Kounoike Hospital, Nara 639-2273, Japan; 2Hirakawa Hospital, Tokyo 192-0152, Japan; 3Maplehill Hospital, Hiroshima 739-0651, Japan; 4Faculty of Health Science, Department of Rehabilitation, Hiroshima Cosmopolitan University, Hiroshima 731-3166, Japan; 5Department of Rehabilitation Medicine, Iwate Medical University, Shiwa-gun 028-3694, Japan; 6Department of Rehabilitation Medicine, Wakayama Medical University, Wakayama 641-8509, Japan; 7Rehabilitation Center, Faculty of Medicine, University of Miyazaki, Miyazaki 889-1692, Japan

**Keywords:** motor function, physical activity, psychiatric inpatients, physical therapist

## Abstract

The aim of this cross-sectional study was to determine the status of locomotive syndrome (LS) and the level of physical activity (PA) in long-term inpatients in a psychiatric care ward and to investigate the association between the severity of LS and the level of PA. The study participants consisted of 25 patients aged 55 years or older who had been admitted to a psychiatric care ward for more than one year. The participants’ LS stage was determined and their level of PA was measured using an accelerometer. We also analyzed the correlations between the LS stage test results, level of PA, and values for each assessment item. The LS stage test showed that 84.0% of the participants were at stages 3. The participants’ mean step count was 3089.8 ± 2346.5 steps. The participants’ mean sedentary time was 349.7 ± 68.9 min, which is more than 70% of the total measuring time. Overall, the results indicate that LS stage was significantly correlated to age, ADL, and level of PA. Patients who stay in a psychiatric care ward experience declining motor functioning and lack PA. Deterioration of motor functioning is associated with lack of PA, suggesting the need for physical intervention.

## 1. Introduction

In Japan, aging and long-term hospitalization pose serious problems for psychiatric care inpatients. In 2017, 16.2%, 23.8%, and 38.1% of psychiatric care inpatients were 55–64, 65–74, and >75 years of age, respectively [1]. In Japan, the average length of stay (LOS) for the treatment of psychiatric and behavioral impairments is 277.1 days [1]. Moreover, long periods of hospitalization are common in psychiatric care wards; 51.1% and 82.2% of inpatients have LOSs that exceed 5 and 1 year, respectively [2]. Older and longer-term psychiatric inpatients are likely in poor physical health and they tend to develop more serious complications, including physical disability and falling incidents.

Over 80 years ago, the BMJ reported that mental illness was associated with poor physical health [3]. Subsequent studies have shown that the risk for premature mortality of people with severe mental illness is 2–3 times higher than that of the general population [4], with cardiovascular disease causing most of the early mortalities [5]. In the general population, there is ample evidence demonstrating that low cardiorespiratory fitness is a strong and independent predictor of cardiovascular disease and all-cause mortality [6]. A meta-analysis of published works indicates that cardiorespiratory fitness in people with severe mental illness is severely reduced compared to age- and sex-matched controls [7]. Thus, physical fitness is a major factor determining the health of patients with mental illness. The following factors associated with low physical fitness have been found in patients with schizophrenia: lack of physical activity (PA), illness duration, smoking, metabolic syndrome, and the presence of more severe negative, depressive, and cognitive symptoms [8]; and patients with bipolar disorder are often associated with older age, lack of PA, positive and negative affect, and consumption of antipsychotic medication [9].

Only a few studies have investigated the decline in physical fitness and the factors contributing to such decline in elderly, long-term hospitalized psychiatric patients. In our previous study [10], we investigated the motor functioning of long-term psychiatric Japanese inpatients who were suffering from the locomotive syndrome (LS), taking into account the characteristics associated with aging. The Japanese Orthopaedic Association (JOA) defines LS as a condition in which the mobility function of the patient is impaired due to musculoskeletal disorders, and as it progresses, patients tend to require more nursing care [11]. Our previous study found that an overwhelming majority of long-stay psychiatric care inpatients suffer from declining motor function due to aging and long-term hospitalization, consequently affecting their activities of daily living (ADL). Psychiatric care inpatients often have fixed schedules for treatment and daily living. However, hospital life tends to involve low levels of PA, which may be associated with a decline in physical fitness. Therefore, PA is an important consideration for psychiatric care inpatients; however, in our previous studies, we have not objectively measured PA.

There is growing evidence that patients with mental illness undergo a decrease in PA [12,13,14,15,16], which is a serious problem because it may be associated with cardiovascular disease, premature mortality, and a decline in physical fitness.

Therefore, the purpose of this study was to determine the status of LS and the level of PA in long-term inpatients in a psychiatric care ward and to examine the association between the severity of LS and the level of PA.

## 2. Materials and Methods

### 2.1. Study Participants

This is a single-center study. The subjects consisted of 49 patients aged 55 years or older who had been admitted to psychiatric care wards for more than one year. Sixteen patients were unable to walk and seven did not give their informed consent and were excluded from the study. Finally, 25 patients participated in this study.

### 2.2. Basic Information

We obtained the following information from the participants’ medical records: age, sex, F code in the International Classification of Diseases 10th edition (ICD-10), LOS, body mass index (BMI), and chlorpromazine-equivalent dose of antipsychotics (CP dose) [17]. Physical therapists evaluated each of the participant’s independence level in ADL using the Functional Independence Measure (FIM) [18]; mental function was assessed with the Global Assessment of Functioning (GAF) [19]; cognitive function was assessed with the Japanese version of the Montreal Cognitive Assessment (MoCA-J) [20].

### 2.3. Physical Activity and Sedentary Behavior

All participants were fitted with a triaxial accelerometer (Panasonic; Acti-marker EW4800, Kadoma, Japan), which they wore for 2 days from 9:00 to 17:00. PA was measured in steps taken and expressed as metabolic equivalents (MET; in kcal/h/kg), an indicator of daily energy expenditure [21]. Based on the model proposed by Pate et al. [22] for classifying the MET intensity of PAs, PA was classified as sedentary, 1.0–1.5 METs; light (LPA), 1.6–3.0 METs; moderate-to-vigorous (MVPA), >3.0 METs. Daily average times spent in sedentary, LPA, and MVPA were calculated.

### 2.4. Locomotive Syndrome Stage Tests

To assess the LS risk, we administered a stand-up test, a two-step test, and the 25-item Geriatric Locomotive Function Scale (GLFS-25) test, as described below. Based on clinical levels developed by the JOA [23], the participants were classified as having no LS risk or having LS risk levels of 1 (LS stage 1), 2 (LS stage 2), or 3 (LS stage 3). Descriptions of these stages are given below.

In the stand-up test, physical therapists measured the participants’ ability to adopt a single-leg or double-leg stance from stools that were 40, 30, 20, and 10 cm high. The result was reported as the minimum stool height from which the participant could stand.

In the two-step test, the physical therapists measured the two-stride length of each participant, starting from a starting line to the final position of the tips of the toes. The score was calculated by dividing the maximal length of two steps by the participant’s height.

The GLFS-25 is a self-reported, comprehensive survey limited to information relevant to the preceding month and included 4 items on pain, 16 items on ADL, 3 items on social functions, and 2 items on mental health status. Each item was graded from no impairment (0 points) to severe impairment (4 points), and thus total scores ranged from 0 (no symptoms) to 100 (most severe symptoms) points.

The criteria for LS stage 1 were any of the following: (1) difficulty rising on 1 leg (either leg) from a 40-cm-high stool in the stand-up test; (2) a two-step test score of <1.3; and (3) a GLFS-25 score of ≥7. Any of these conditions resulted in a diagnosis of LS stage 1, indicating the beginning of a decline in mobility. The criteria for LS stage 2 were as follows: (1) difficulty rising on both legs from a 20-cm-high stool in the stand-up test; (2) a two-step test score of <1.1; and (3) a GLFS-25 score of Th16. Any participant who met these conditions was diagnosed with LS stage 2, indicating progression toward decline in mobility that increases the risk of losing the ability to live independently. The criteria for LS stage 3 were as follows: (1) difficulty rising on both legs from a 30-cm-high stool in the stand-up test; (2) a two-step test score of <0.9; and (3) a GLFS-25 score of ≥24. Any participant who met these conditions was diagnosed with LS stage 3, indicating a progressive decline in mobility that interferes with social participation.

### 2.5. Statistical Analysis

Basic statistics for each survey item and correlation analysis were performed using statistical analysis software R, version 4.0.2 (The R Foundation, Boston, MA, USA). Spearman’s rank correlations were analyzed between the LS stage tests (stand-up test, two-step test, and GLFS-25) and age, LOS, BMI, CP dose, FIM score, GAF score, MoCA-J score, and PA (step counts, LPA, MVPA, sedentary time). Correlations were considered significant at a *p*-value of 5%.

## 3. Results

### 3.1. Characteristics of Study Participants

Table 1 lists the mean values of age, LOS, BMI, and CP dose of the 25 participants included in the final analysis. The participants’ mean age was 70.1 ± 9.3 years, with ages ranging from 55 to 92 years; 80.0% of the participants were aged 65 years or older. The participants’ mean LOS was 16.2 ± 14.1 years. Over half of the participants had stayed in the psychiatric ward for >10 years. The participants’ mean CP dose was 646.9 ± 741.6 mg. We defined a high CP dose regimen as a prescription of >1000 chlorpromazine-equivalent mg of antipsychotics per day, a regimen that 16.0% of the participants received. Based on ICD-10 F code sorting, 92.0% of all participants were diagnosed as either F2 or F3 (Table 2). Table 3 lists the mean FIM and GAF scores, as well as the means of other measured variables. The participants’ mean scores for FIM motor and cognitive items were 82.0 ± 9.1 and 27.4 ± 5.5 points, respectively. More than 90% of the participants’ GAF scores were below 40 (GAF scores below 40 had a higher probability of readmission to the hospital [19]). The participants’ mean MoCA-J score was 13.9 ± 6.8 points, with all scoring below 25 points, the cut-off value for mild cognitive impairment.

### 3.2. Physical Activity Levels of the Participants

Table 4 lists the mean values of step counts, sedentary time, LPA time, and MVPA time. The participants’ mean step count was 3089.8 ± 2346.5 steps, with all the participants scoring below 8000 steps per day, the recommended value for physical health [24,25,26,27]. The participants’ mean sedentary time was 349.7 ± 68.9 min, which is more than 70% of the time between 9:00 to 17:00.

### 3.3. Distribution of Risk Levels for Locomotive Syndrome

Figure 1 shows the distribution of LS risk levels based on the results of the stand-up, two-step, and GLFS-25 risk assessment tests and the total assessment for all participants. In the total assessment, all of them were determined to be LS (stages 1, 2, and 3).

### 3.4. Association between Locomotive Syndrome and Measured Variables

Table 5 shows the correlations between the scores of the stand-up, two-step, and GLFS-25 tests and the measured variables. The stand-up test scores were significantly negatively correlated with age (r = −0.65, *p* = 0.02), and significantly positively correlated with the FIM motor items score (r = 0.78, *p* < 0.01) and step counts (r = 0.68, *p* < 0.01). The two-step test scores were significantly positively correlated with the FIM motor items score (r = 0.73, *p* < 0.01) and step counts (r = 0.27, *p* = 0.05). The GLFS-25 scores were significantly positively correlated with age (r = 0.65, *p* < 0.01) and significantly negatively correlated with the FIM motor items score (r = −0.68, *p* < 0.01).

## 4. Discussion

The purpose of this study was to determine the status of LS and PA in long-term inpatients in a psychiatric care ward and to investigate the association between the severity of LS and the level of PA. Our results show that most of the long-term inpatients in the psychiatric ward were at LS stages 2 or 3, and that the levels of PA in these patients were also declining. The severity of LS was related not only to age and ADL, but also to the level of PA, suggesting the need for physical intervention. The novelty of this study is that it focused on long-term hospitalization and aged patients, which characterize psychiatric care patients in Japan. We evaluated the motor functioning and measured the level of PA in elderly patients who had been admitted to psychiatric care wards for long periods of time. In addition, previous studies have often measured the level of PA subjectively, such as by self-reporting by questionnaires; however, compared to objective measurements, this method underestimates sedentary behavior while overestimating MVPA [12,13,14,15,16]. Therefore, we believe that the use of an accelerometer in this study was appropriate and provided an objective measure of PA.

The motor functioning of patients with mental illness is often impaired. In addition, the risk for premature mortality in people with severe mental illness is 2–3 times higher than that of the general population [4]. For example, schizophrenia is associated with accelerated aging syndrome, in which the physiological changes in body structures and functions that are associated with normal aging occur approximately 25 years earlier [28]. In the present study, 84.0% of the participants had the most severe form of LS (stage 3), confirming that motor functioning declines in long-term inpatients in psychiatric care wards. The prevalence of LS risk increases with age, as does the prevalence of age-related decline in motor function. Yoshimura et al. [29] estimated the prevalence of LS risk by age, based on the results of LS grade tests administered to the general population of a community. The prevalence of LS stage 2 was 28.2% and 39.0% in men and women in their 70s, respectively; and 62.1% and 76.0% in men and women in their 80s and over, respectively. Although the average age of the participants in the present study was 70 years old, all were diagnosed with LS, with 84.0% and 12.0% of them having LS stage 3 and 2, respectively. This indicates that inpatients in psychiatric care wards are at a higher risk of LS than members of the same age in the general population.

In a previous study, Aoyagi et al. reported that elderly members of a community performed an average of 6500 steps per day and an average of 17 min of moderate- to high-intensity activities [24]. In the present study, the participants per day and an av steps taken per day was 3089, and their average MVPA duration was 9 min, suggesting a lack of PA, although it is necessary to take into account that the measurement period of the present study was from 9:00 to 17:00. Aoyagi et al. established that physical health requires at least 8000 steps of activity per day [25,26,27]. None of the participants in the present study exceeded 8000 steps, thus, they did not reach the level of activity needed to maintain physical health. Moreover, the average duration of sedentary behavior in this study was 350 min, which represents more than 70% of the total measuring time. Several meta-analyses on the level of PA in mentally ill patients agree that compared to the general population, mentally ill patients spend significantly more time sedentary [12,13,14,15,16], averaging 660 min per day [14]. This corresponds to 60–70% of awake time, a level comparable to the results of the present study. These results indicate that patients admitted to psychiatric care wards spend long periods sedentary.

The current study showed that step counts, as a primary physical activity index, were correlated with the scores in stand-up and two-step tests but not with GLFS-25. The participants who had been admitted to a psychiatric care ward for >1 year showed low GAF and MOCA-J scores (e.g., negative symptoms, depressive symptoms, and cognitive function), which may have affected the results of the GLFS-25, a self-reported questionnaire. The correlation between PA and LS was only observed in step counts but not in sedentary time, LPA, and MVPA. The participants could walk of their own free will in the psychiatric care wards; however, most participants led a low-stimulation, low-activity hospital life due to the fixed daily treatment and living schedule. This suggests that the environment may have affected the MET intensity of PAs. In the present study, we found an association between the LS stage test results and PA, which is consistent with many previous studies that have found an association between motor functioning and PA in patients with mental illness [8,9,30]. Psychiatric inpatients are deficient in PA, and the fact that LS risk is associated with PA suggests that efforts to increase PA may prevent the development of metabolic syndrome and decreased health-related quality of life (QOL) [31,32,33]. Previous studies have also reported that PA is beneficial for mental health and QOL, and it helps alleviate psychiatric symptoms in psychiatric patients [34,35,36]. Factors contributing to inactivity in psychiatric inpatients include their daily fixed schedule that lacks time PA, their limited living space, the effects of antipsychotic medication, and psychiatric symptoms (such as negative, depressive, and cognitive symptoms) that hinder engaging in PA. Physical therapists can contribute to the solution of these problems by increasing their understanding of the characteristics of schizophrenia and other mental and behavioral disorders getting involved in the daily activities of patients, improving physical mobility, and expanding the living space. Moreover, other interventions can also be considered, such as establishing mobility patterns and living environments that allow patients to move safely and comfortably, conducting group exercise programs developed by psychiatric occupational therapists, and cooperating with ward staff in creating care plans.

This study has a number of limitations. First, we conducted a single-center study involving a small number of subjects. It will be necessary to conduct a multi-center study involving a larger number of subjects to determine whether our results are consistent across other psychiatric wards. Second, this was a cross-sectional study that only showed the association between LS risk and PA. Therefore, longitudinal studies that clarify causal relationships are needed, along with studies that examine the effects of efforts aiming to prevent the decline of motor functioning, including LS, in psychiatric inpatients. Finally, Kota et al. [37] reported that among patients admitted to a psychiatric ward, LS risk as judged by the two-step test is associated with the decision of discharge to the community. We believe that it is necessary to investigate whether improving motor functioning can facilitate patient discharge. Our results indicate that a decline in motor functioning of psychiatric inpatients is associated with a lack of PA, suggesting an urgent need for physical rehabilitation. We recommend that physical therapists play a larger role in inpatient intervention in psychiatric care wards.

## 5. Conclusions

Patients who stayed in a psychiatric care ward experience declining motor function and lack PA. Outcomes of the present study revealed that 84.0% of the participants had the most severe form of LS (stage 3), which confirmed a long-term decline in motor functioning among inpatients in psychiatric care wards. The average number of steps taken by the participants per day was 3089, whereas their average MVPA duration was 9 min, thereby suggesting a lack of PA. Deterioration of motor function was associated with a lack of PA, and indicated the need for physical intervention. We recommend that physical therapists should have a larger role in the intervention of psychiatric care ward inpatients.

## Figures and Tables

**Figure 1 healthcare-10-01741-f001:**
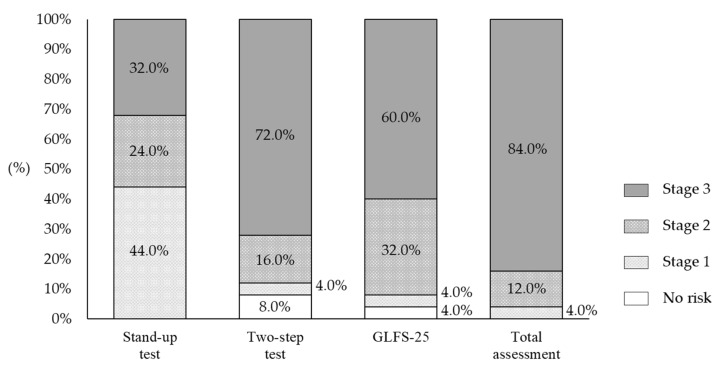
Distribution of risk levels for locomotive syndrome based on the results of the stand-up test, two-step test, and GLFS-25.

**Table 1 healthcare-10-01741-t001:** Characteristics of the study participants.

Variables	Overall (*n* = 25)	Male (*n* = 5)	Female (*n* = 20)
Age, y	70.1 ± 9.3	67.0 ± 7.6	70.9 ± 9.7
≤59 y	5 (20.0%)	1 (20.0%)	4 (20.0%)
60–69 y	7 (28.0%)	2 (40.0%)	5 (25.0%)
70–79 y	9 (36.0%)	2 (40.0%)	7 (35.0%)
≥80 y	4 (16.0%)	0 (0.0%)	4 (20.0%)
LOS, y	16.2 ± 14.1	17.5 ± 16.9	15.8 ± 13.8
≥1y, <5y	6 (24.0%)	2 (40.0%)	4 (20.0%)
≥5y, <10y	6 (24.0%)	0 (0.0%)	6 (30.0%)
≥10y	13 (52.0%)	3 (60.0%)	10 (50.0%)
BMI (kg/m^2^)	22.5 ± 3.9	20.9 ± 4.1	22.9 ± 3.9
CP dose (mg/day)	646.9 ± 741.6	745.9 ± 611.3	621.1 ± 763.0

Continuous variables are expressed as mean ± SD. Categorical variables are expressed as n (%). LOS, length of stay; BMI, body mass index; CP dose, chlorpromazine equivalent dose of antipsychotics.

**Table 2 healthcare-10-01741-t002:** Diagnoses (F-codes) of all participants based on the ICD-10 classification of mental and behavioral disorders.

Categories	Overall (*n* = 25)	Male (*n* = 5)	Female (*n* = 20)
F0	1 (4.0%)	0 (0.0%)	1 (5.0%)
F2	18 (72.0%)	3 (60.0%)	15 (75.0%)
F3	5 (20.0%)	2 (40.0%)	3 (15.0%)
F7	1 (4.0%)	0 (0.0%)	1 (5.0%)

Categorical variables are expressed as n (%). F0, Organic, including symptomatic, mental disorders; F2, Schizophrenia, schizotypal and delusional disorders; F3, Mood [affective] disorders; F7, Mental retardation; ICD-10, the International Classification of Diseases 10th edition.

**Table 3 healthcare-10-01741-t003:** Independence level in ADL, mental function, and cognitive function of the participants.

Variables		Overall (*n* = 25)	Male (*n* = 5)	Female (*n* = 20)
FIM-motor		82.0 ± 9.1	83.8 ± 10.5	81.6 ± 8.9
FIM-cognitive		27.4 ± 5.5	27.6 ± 7.5	27.4 ± 5.2
GAF	11–20	1 (4.0%)	1 (20.0%)	0 (0.0%)
	21–30	16 (64.0%)	3 (60.0%)	13 (65.0%)
	31–40	6 (24.0%)	1 (20.0%)	5 (25.0%)
	41–50	1 (4.0%)	0 (0.0%)	1 (5.0%)
	51–60	1 (4.0%)	0 (0.0%)	1 (5.0%)
MoCA-J		13.9 ± 6.8	15.0 ± 9.2	13.6 ± 6.6

Continuous variables are expressed as mean ± SD. Categorical variables are expressed as n (%). FIM, Functional Independence Measure; GAF, Global Assessment of Functioning; MoCA-J, Japanese version of The Montreal Cognitive Assessment.

**Table 4 healthcare-10-01741-t004:** Physical activity levels of the participants measured from 09:00 to 17:00.

Variables	Overall (*n* = 25)	Male (*n* = 5)	Female (*n* = 20)
Step counts (steps)	3089.8 ± 2346.5	4243.3 ± 1685.3	2955.5 ± 2450.1
Sedentary time (min)	349.7 ± 68.6	366.3 ± 34.7	342.1 ± 74.7
LPA time (min)	122.4 ± 66.9	107.0 ± 32.1	129.8 ± 72.8
MVPA time (min)	9.2 ± 7.7	8.0± 9.2	9.3 ± 7.9

Continuous variables are expressed as mean ± SD. LPA, Light-intensity physical activity; MVPA, Moderate-to-vigorous physical activity.

**Table 5 healthcare-10-01741-t005:** Correlations between the stand-up test, two-step test, and GLFS-25 scores and measured variables in participants.

Variables	Stand-Up Test	Two-Step Test	GLFS-25
Age	−0.65 *	−0.59	0.65 **
LOS	0.02	−0.12	0.27
BMI	0.42	0.30	−0.33
CP dose	0.18	0.42	−0.21
FIM-motor	0.78 **	0.73 **	−0.68 **
FIM-cognitive	0.44	0.49	−0.32
GAF	0.31	0.25	−0.19
MoCA-J	0.21	−0.06	−0.15
Step counts	0.68 **	0.27 *	−0.41
Sedentary time	−0.21	0.11	0.28
LPA	0.22	−0.06	−0.30
MVPA	0.35	0.11	−0.36

GLFS-25, 25-Geriatric Locomotive Function Scale; LOS, Length of stay; BMI, Body Mass Index; CP dose, Chlorpromazine equivalent dose of antipsychotics; FIM, Functional Independence Measure; GAF, Global Assessment of Functioning; MoCA-J, Japanese version of The Montreal Cognitive Assessment; LPA, Light-intensity physical activity; MVPA, Moderate-to-vigorous physical activity. Data are Spearman’s rank correlation coefficients (r). * *p* < 0.05, ** *p* < 0.01.

## Data Availability

Not applicable.

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
