# Peer review of "Association between Locomotive Syndrome and Physical Activity in Long-Term Inpatients of Psychiatric Care Wards in Japan: A Preliminary Study"

_healthcare, 2022, doi:10.3390/healthcare10091741_

Round 1

Reviewer 1 Report

The work is original and is structured correctly.

The introduction, despite not being extensive, does find the most important references in relation to the study. The materials and methods have been clearly exposed as well as the results. At the level of discussion I find it correctly referenced.

It would have to contribute that the conclusions could be improved since it does not make much reference to the results and the extension is short.

1. What is the main question that the research addresses?
See the relationship between locomotor syndrome and physical activity in patients admitted to a psychiatric hospital
2. Do you consider the topic original or relevant in the field, and if
then why?
It is original given that physical activity is being seen to have a positive influence on health, so researching in patients hospitalized for mental problems and who do not have a relationship with normal society is a novelty, to see how it influences their state of health.
3. What does it contribute to the subject area compared to other publications?
material?
There are several published studies, but carrying out a field study in a boarding school and relating it is not enough.
4. What specific improvements could the authors consider regarding the
methodology?
Regarding the methodology I see it correct
5. Are the references appropriate? If the references sought are appropriate for the topic they develop

Reviewer 2 Report

The study examines the relationship between locomotive syndrome and physical activity in long-term inpatients of psychiatric care wards, showing the correlation between locomotive syndrome and physical activity. Overall, I found that the manuscript is well-written and the issue is clear, and the results are interesting. I only have some minor comments.

The authors previously examined the link between physical functioning and locomotive syndrome in the same cohort with taking an “age” factor into account [ref 10; line 58]. The current results show a strong correlation between physical activity and age (Table 5), however, the authors did not take “age” into account. As age also correlates with locomotive syndrome level, I suggest that the authors could use regression or other methods to take “age” into account to examine whether the correlation between physical functioning and locomotive syndrome is still pronounced.

Another comment is that wording of many concepts is hard to differentiate. In particular, I found that physical functioning seems to be different from physical activity. However, it is hard to understand what is “physical functioning”. I would thus suggest the authors make a clear definition of some similar concepts.

Steps counts, as a primary physical activity index, correlate with the scores in stand-up and two-step tests, but not with GLFS-25. It would be great if the authors can elaborate why steps counts correlate with some LS indices but not others.

Similarly, the correlation between physical activity and locomotive syndrome was only observed in steps counts, but, not in other physical activity indices (e.g., sedentary time, LPA, MVPA). What do these results mean? If some other indices are computed according to the model proposed by Pate, do these results suggest that other indices according to the model are less sensitive to indicate physical activity level?
